# Recurrent processes support a cascade of hierarchical decisions

Laura Gwilliams[1,2]*, Jean-Remi King[1,3,4]

[1]Department of Psychology, New York University, New York, United States; [2]NYU Abu Dhabi Institute, Abu Dhabi, United Arab Emirates; [3]Frankfurt Institute for Advanced Studies, Frankfurt, Germany; [4]Laboratoire des Systèmes Perceptifs (CNRS UMR 8248), Département d'Études Cognitives, École Normale Supérieure, PSL University, Paris, France

**Abstract** Perception depends on a complex interplay between feedforward and recurrent processing. Yet, while the former has been extensively characterized, the computational organization of the latter remains largely unknown. Here, we use magneto-encephalography to localize, track and decode the feedforward and recurrent processes of reading, as elicited by letters and digits whose level of ambiguity was parametrically manipulated. We first confirm that a feedforward response propagates through the ventral and dorsal pathways within the first 200 ms. The subsequent activity is distributed across temporal, parietal and prefrontal cortices, which sequentially generate five levels of representations culminating in action-specific motor signals. Our decoding analyses reveal that both the content and the timing of these brain responses are best explained by a hierarchy of recurrent neural assemblies, which both maintain and broadcast increasingly rich representations. Together, these results show how recurrent processes generate, over extended time periods, a cascade of decisions that ultimately accounts for subjects' perceptual reports and reaction times.

*For correspondence:
leg5@nyu.edu

**Competing interests:** The authors declare that no competing interests exist.

## Introduction

To process the rich sensory flow emanating from the retina, the brain recruits a hierarchical network originating in the primary visual areas and culminating in the infero-temporal, dorso-parietal and pre-frontal cortices (*Hubel and Wiesel, 1962*; *Maunsell and van Essen, 1983*; *Riesenhuber and Poggio, 1999*; *DiCarlo et al., 2012*).

In theory, the feedforward recruitment of this neural hierarchy could suffice to explain our ability to recognize visual objects. For example, recent studies demonstrate that artificial feedforward neural networks trained to categorize objects generate similar activations patterns to those elicited in the infero-temporal cortices (*Yamins et al., 2014*; *Schrimpf et al., 2019*; *Khaligh-Razavi and Kriegeskorte, 2014*; *Cichy et al., 2016*).

However, feedforward architectures have a fixed number of processing stages and are thus unable to explain a number of neural and perceptual phenomena. For example, the time it takes subjects to recognize objects considerably varies from one trial to the next (*Ratcliff and Smith, 2004*). In addition, the neural responses to visual stimuli generally exceed the 200 ms feedforward recruitment of the visual hierarchy (*Dehaene and Changeux, 2011a*; *Lamme and Roelfsema, 2000*).

A large body of research shows that recurrent processing accounts for such behavioral and neural dynamics (*Lamme and Roelfsema, 2000*; *Gray et al., 1989*; *Gold and Shadlen, 2007*; *Shadlen and Newsome, 2001*; *O'Connell et al., 2012*; *Spoerer et al., 2017*; *Kar et al., 2019*; *Kietzmann et al., 2019*; *Spoerer et al., 2019*). In this view, recurrent processing would mainly consist of accumulating sensory evidence until a decision to act is triggered (*O'Connell et al., 2012*; *Mohsenzadeh et al., 2018*; *Rajaei et al., 2019*).

However, the precise neuronal and computational organization of recurrent processing remains unclear at the system level. In particular, how distinct recurrent assemblies implement series of hierarchical decisions remains a major unknown.

To address this issue, we use magneto-encephalography (MEG) and structural magnetic-resonance imaging (MRI) to localize, track and decode, from whole-brain activity, the feedforward (0–200 ms) and recurrent processes (>200 ms) elicited by variably ambiguous characters briefly flashed on a computer screen. We show that the late and sustained neural activity distributed along the visual pathways generates, over extended time periods, a cascade of categorical decisions that ultimately predicts subjects' perceptual reports.

## Results

### Subjective reports of stimulus identity are categorical

To investigate the brain and computational bases of perceptual recognition, we used visual characters as described in *King and Dehaene, 2014a*. These stimuli can be parametrically morphed between specific letters and digits by varying the contrast of their individual edges, hereafter referred to as pixels (*Figure 1A–B*).

To check that these stimuli create categorical percepts, we asked eight human subjects to provide continuous subjective reports by clicking on a disk after each stimulus presentation (Experiment 1. *Figure 1A*). The radius and the angle of the response on this disk indicated the subjective visibility and the subjective identity of the stimulus, respectively. We then compared (i) the reported angle with (ii) the stimulus evidence (i.e. the expected angle given the pixels) for each morph separately (e.g. 5–6, 6–8, etc). Subjective reports were categorical: cross-validated sigmoidal models better predicted subjects' responses (r = 0.49 ± 0.05, p=0.002) than linear models (r = 0.46 ± 0.043, p=0.002, sigmoid > linear: p=0.017 *Figure 1B–C*).

We adapted this experimental paradigm for an MEG experiment by modifying three main aspects (Experiment 2). First, we used stimuli that could be morphed between letters and digits, to trigger macroscopically distinguishable brain responses in the visual word form area (VWFA) and number form area (NFA) (*Dehaene and Cohen, 2011b*; *Shum et al., 2013*). Second, we added two task-irrelevant flankers next to the target stimulus (*Figure 1D*) to increase our chances of eliciting recurrent processes via crowding (*Strasburger et al., 2011*; *Pelli et al., 2004*). Note that this assumption is based on previous work; we did not test the effect of crowding explicitly in this study. Third, a new set of 17 subjects reported subjective identity via a two-alternative forced-choice button press. The identity-response mapping was orthogonal to the letter/digit category and changed on every block of 48 trials. There were 1920 trials total, 320 of which were presented passively, were not ambiguous and did not require a response.

Perceptual reports followed a similar sigmoidal pattern to Experiment 1: performance was worse for more ambiguous trials (65%) as compared to unambiguous trials (92%, p<0.001). In addition, reaction time slightly, and consistently, increased with uncertainty (i.e. how ambiguous the stimulus is). For example, highly ambiguous stimuli were identified within 690 ms, whereas nonambiguous stimuli were identified within 624 ms (z = −21.68, p<0.001) (*Figure 1E–F*). Although subjects were asked to respond as quickly as possible, the observed reaction times were overall quite slow, reflecting the difficulty of the task.

### Neural representations are functionally organized over time and space

Here, we aimed to decompose the sequence of decisions that allow subjects to transform raw visual input into perceptual reports. To this aim, we localized the MEG signals onto subjects' structural MRI with dynamic statistical parametric mapping (dSPM, *Dale et al., 2000*), and morphed these source estimates onto a common brain coordinate (*Fischl, 2012*; *Gramfort et al., 2014*). The results confirmed that the stimuli elicited, on average, a sharp response in the primary visual areas around 70 ms, followed by a fast feedforward response along the ventral and dorsal visual pathways within the first 150–200 ms. After 200 ms, the activity appeared sustained and widely distributed across the associative cortices up until 500–600 ms after stimulus onset (*Figure 1G* and *Video 1*).

To separate the processing stages underlying these neural responses, we applied (i) mass-univariate; (ii) temporal decoding and (iii) spatial decoding analyses based on the five orthogonal features

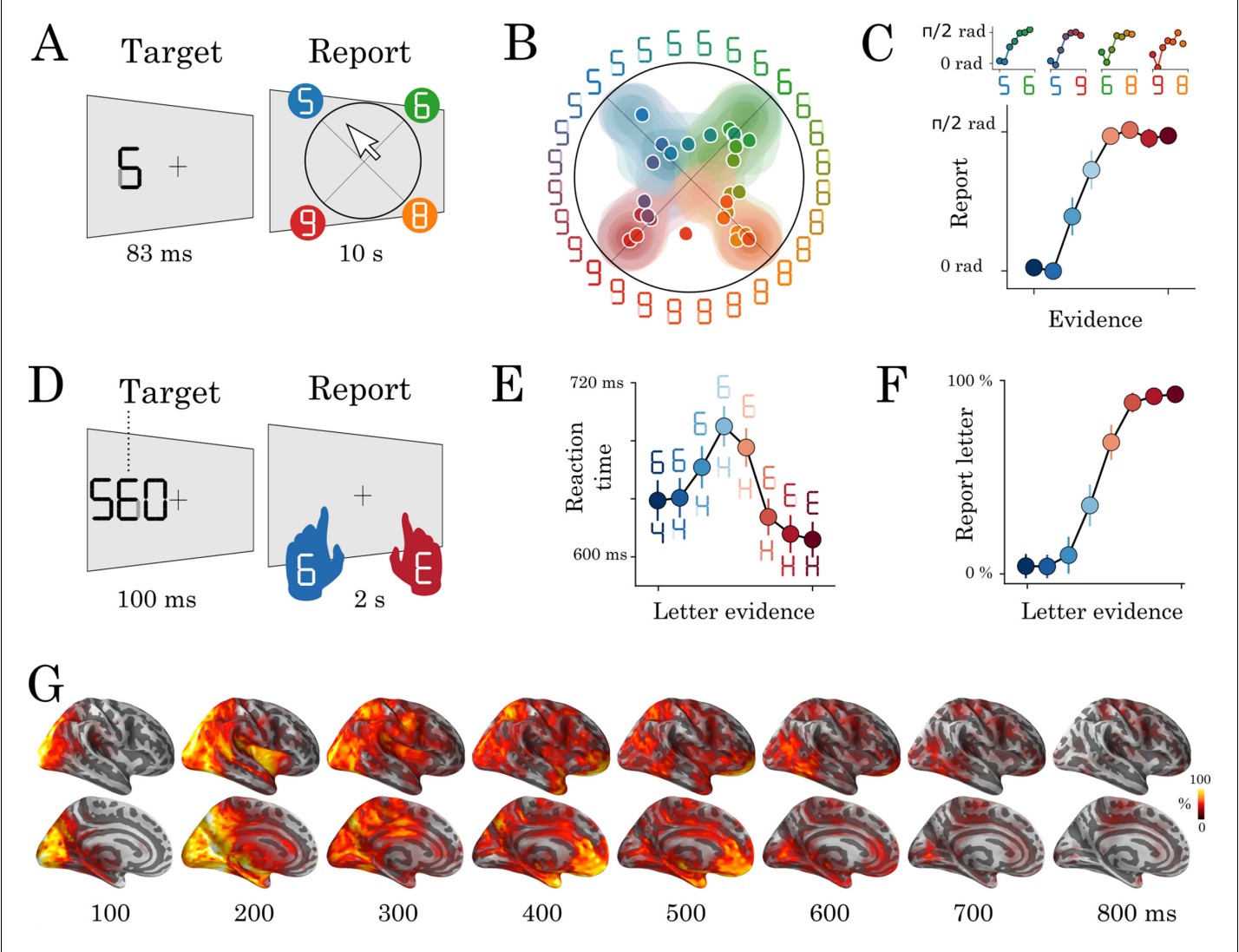

**Figure 1.** Experimental protocol and behavioral results. Experiment 1: eight human subjects provided perceptual judgments on variably ambiguous digits briefly flashed at the center of a computer screen (A). Reports were made by clicking on a disk, where (i) the radius and (ii) the angle on the disk indicate (i) subjective visibility and (ii) subjective identity respectively. (B) Distribution (areas) and mean response (dots) location for each color-coded stimulus. (C) Top plots show the same data as B, broken down for each morph set. The x-axis indicates the expected angle given the stimulus pixels (color-coded), hereafter referred to as evidence. The y-axis indicates the angle of the mean response relative to stimulus evidence. The bottom plot shows the same data, grouped across morphs. (D) Experiment 2: seventeen subjects categorized a briefly flashed and parametrically manipulated-morph using a two-alternative forced-choice. Stimulus-response mapping changed on every block. (E) Mean reaction times as a function of categorical evidence (the extent to which the stimulus objectively corresponds to a letter). (F) Mean probability of reporting a letter as a function of categorical evidence. (G) Evoked activity estimated with dSPM and estimated across all trials and all subjects. These data are also displayed in *Video 1*. Error-bars indicate the standard-error-of-the-mean (SEM) across subjects.

varying in our study: (1) the position of the stimulus, (2) its identity, (3) its perceived category, (4) its uncertainty and (5) its corresponding button press.

## Mass-univariate

First, we modeled the source localized neural responses over time and space as a function of the five predictors of interest (multivariate in feature space, univariate in source space). Regressor beta coefficients were estimated for each subject separately, then submitted to a spatio-temporal cluster test (*Figure 2A*, details provided in Materials and methods). For stimulus position two clusters were found. One in the left hemisphere (number of sources = 2562, mean t-value = 2.87, 20–1560 ms,

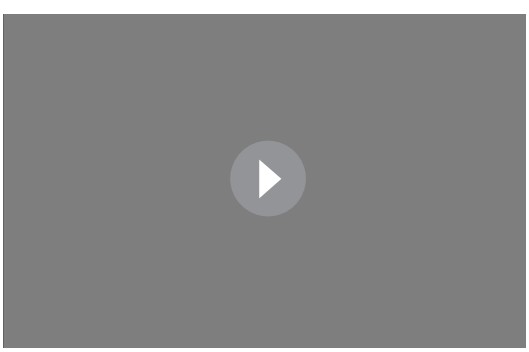

**Video 1.** Source-localized evoked response averaged over all trials and subjects. Activity is plot in noise-normalized dSPM units, and shown on an inflated cortical surface (center) as well as a two-dimensional 'glass brain' that shows activity averaged over the transverse plane (bottom right).
https://elifesciences.org/articles/56603#video1

p=0.001); and one in the right (number of sources = 2562, mean t-value = 2.92, 40–1560 ms, p=0.0005). Stimulus identity also elicited two clusters. One in the left hemisphere (number of sources = 1946, mean t-value = 2.54, 100–840 ms, p=0.0005); and one in the right (number of sources = 2118, mean t-value = 2.48, 120–860 ms, p=0.001). No significant clusters were found for decision, but the largest cluster ranged from 210 to 320 ms (mean t-value = 1.79, p=0.21). Uncertainty elicited two clusters. One in the left hemisphere (number of sources = 2485, mean t-value = 2.42, 280–1560, p=0.001); and one in the right (number of sources = 2319, mean t-value = 2.4, 340–1560 ms, p=0.008). Motor side resulted in two clusters. One in the left hemisphere (number of sources = 2523, mean t-value = 2.56, 280–1560, p=0.0005); and one in the right (number of sources = 2525, mean t-value = 2.66, 280–1560 ms, p=0.0005). See *Figure 2—figure supplements 4–8* for a full display of these results.

The perceived category did not yield significant results after correction for multiple comparisons. Thus, the rest of our analyses are based on multivariate techniques (univariate in feature space, multivariate in source space), which provide highly superior statistical sensitivity to our experimental manipulations. The direct comparison between the sensitivity of the mass-univariate versus multivariate approaches are shown in *Figure 2*.

## Temporal and spatial decoding

To overcome the poor SNR of single-trial MEG responses, we next applied multivariate decoding analyses to identify when and where low-level visual features are represented in brain activity. We estimated, at each time sample separately, the ability of an l2-regularized regression to predict, from all MEG sensors, the five stimulus features of interest. Overall, the multivariate analyses were far more sensitive than the univariate tests.

Stimulus position (the location of the stimulus on the computer screen: left versus right) was decodable between 41 and 1500 ms and peaked at 120 ms (AUC = 0.94; SEM = 0.007; p<0.001 as estimated with second-level non-parametric temporal cluster test across subjects, *Figure 2C*). To summarize where stimulus position was represented in the brain, we implemented 'spatial decoders': l2-regularized logistic regressions fit across all time samples (0–1500 ms) for each estimated brain source separately. Spatial decoding peaked in early visual areas and was significant across a large variety of visual and associative cortices as estimated with a second-level non-parametric spatial cluster test across subjects (*Figure 2B*). Stimulus position was encoded in the timecourse of all sources, in both the left hemisphere (mean t-value = 9.42, p=0.0005) and the right (mean t-value = 9.97, p=0.0005). These signals peaked in the early visual cortex (mean MNI [x = 27.59; y = −74.15; z = −1.07]), and propagated along the ventral and dorsal streams during the first 200 ms (*Figure 2A*, *Video 2*); confirming the retinotopic organization of the visual hierarchy (*Hagler and Sereno, 2006*; *Wandell et al., 2007*).

Second, we aimed to isolate more abstract representations related to stimulus identity. Stimulus identity can be analyzed either from an objective referential (what stimulus is objectively presented?) or from a subjective referential (i.e. what stimulus did subjects report having seen?). We first focus on decoding features of the stimulus that are not ambiguous, such that subjective and objective representations are confounded. To this aim, we grouped stimuli along common continua (e.g. The eight stimuli along the 4-H continuum belong to the same morph and are here considered to share a common identity) and fit logistic regression classifiers across morphs (i.e. E-6 versus 4-H). The corresponding stimulus identity was decodable between 120 and 845 ms and peaked at 225 ms

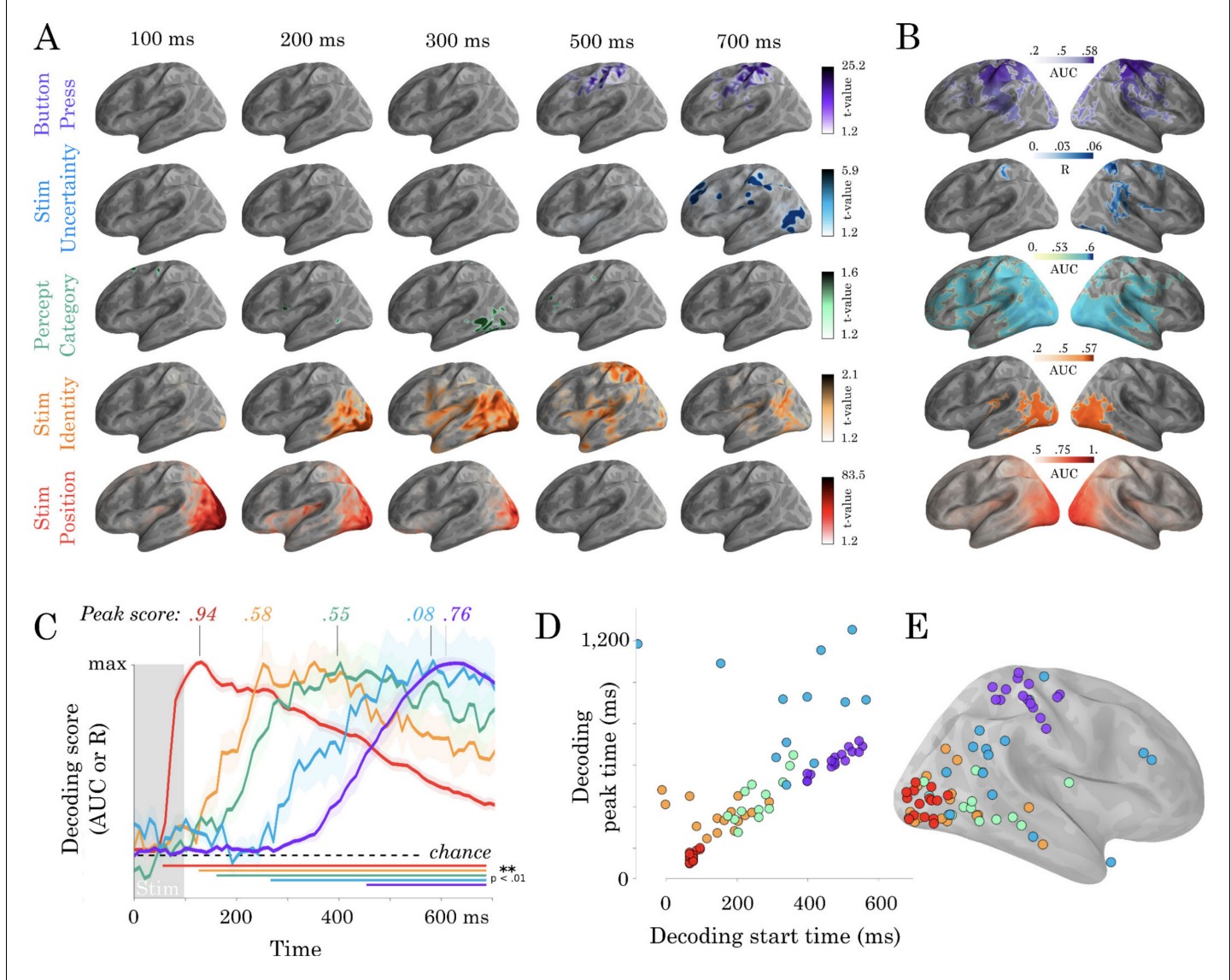

**Figure 2.** Spatio-temporal hierarchy. (**A**) Mass-univariate statistics. Each row plots the average-across-subjects beta coefficients obtained from regression between single-trial evoked activity and each of the five features orthogonally varying in this study. These results are displayed in *Video 2*. Colors are thresholded based on t-values that exceed an uncorrected p<0.1. We chose this threshold because the perceptual category did not exceed the significance threshold in the univariate tests. (**B**) Spatial-decoders, consisting of linear models fit across all time sample for each source separately, summarize where each feature can be decoded. Lines indicate significant clusters of decoding scores across subjects cluster-corrected p<0.05. (**C**) Temporal-decoders, consisting of linear models fit across all MEG channels, for each time sample separately, summarize when each feature can be decoded. To highlight the sequential generation of each representation, decoding scores are normalized by their respective peaks. Additional non-normalized decoding timecourses are available in *Figure 2—figure supplements 1* and *2*. (**D**) The peak and the start of temporal decoding plotted for each subject (dot) and for each feature (color). (**E**) The peak spatial decoding plotted for each subject (dot) and for each feature (color).

The online version of this article includes the following figure supplement(s) for figure 2:

**Figure supplement 1.** Exhaustive set of results for decoding different features of the stimulus, time-locked to stimulus onset.

**Figure supplement 2.** Exhaustive set of results for decoding different features of the stimulus, time-locked to motor onset.

**Figure supplement 3.** Decoding letter/digit contrast for symbols not presented in the active condition, where subjects viewed the symbols passively and did not have to make a motor response.

**Figure supplement 4.** Univariate significance mask for 'Stimulus Position'.

**Figure supplement 5.** Univariate significance mask for 'Stimulus Identity'.

**Figure supplement 6.** Univariate significance mask for 'Uncertainty'.

**Figure supplement 7.** Univariate significance mask for 'Motor Side'.

**Figure supplement 8.** Univariate significance across the four significant features.

*Figure 2 continued on next page*

*Figure 2 continued*

**Figure supplement 9.** Significance of multivariate tests across the five significant features.
**Figure supplement 10.** Non-corrected log-transformed (base 10) p-values for the mass univariate tests, plotted for each of the five features.
**Figure supplement 11.** Average decoding timecourses for each of the five features.
**Figure 2—animation 1.** Violin plot of decoding accuracy for the five features of interest over time.
https://elifesciences.org/articles/56603#fig2video1

(AUC = 0.59; SEM = 0.01; p<0.001). Spatial decoding revealed decodability from all sources (mean t-value = 7.9, p<0.0001). These effects peaked more anteriorly than those of stimulus position (mean MNI: x = 27.75; y = −62.75; z = −1.55; p<0.001).

Third, we aimed to isolate the neural signatures of subjective perceptual categorization and thus focus on decoding ambiguous pixels. To this aim, we grouped stimuli based on whether the subject reported a digit or a letter category. Temporal decoders weakly but significantly classified perceptual category from 150 to 940 ms after stimulus onset and peaked at 370 ms (AUC = 0.55; SEM = 0.01; p<0.001, *Figure 2C*). The corresponding sources also peaked in the inferotemporal cortex but more anteriorly than stimulus identity (x = 30.89; y = −35.64; z = 21.41; p<0.01). As reported above, the mass-univariate effects did not survive correction for multiple comparisons (e.g. 210–320 ms: mean t-value=1.79, p=0.21). Nonetheless, spatial decoders, which mitigate the trade-off between temporal specificity and the necessity to correct statistical estimates for multiple comparisons, showed that perceptual category was reliably decoded from a large set of brain areas (mean t-value=4.82; p<0.001; 594 significant vertices) (*Figure 2B*).

Importantly, when training the classifier on all active trials to distinguish letters (E/H) and digits (4/6), we could significantly (max AUC = 0.55; SEM = 0.011; p<0.01; 200–550 ms) decode this contrast for different unambiguous tokens (A/C versus 9/8) when trials were presented passively (no button press required). The time-course of responses to these passive trials was statistically indistinguishable until 350 ms. Thereafter, active trials lead to significantly higher decoding accuracy than the generalization to passive trials. This suggests that the decoders specifically track the letter/digit representation, independently of pixel arrangements (see *Figure 6—figure supplement 2* for passive decoding performance).

Fourth, trial uncertainty (i.e. the objective distance between the presented stimulus and the closest unambiguous character) could be decoded between 270 and 1485 ms and peaked at 590 ms (l2-regularized Ridge regression fit across sensors, R = 0.12; SEM = 0.024; p<0.01). Uncertainty signals were localized more anteriorly than those of stimulus category (x = 12.58; y = −91.44; z = −1.23; p<0.01). While spatial decoding led to significant clusters in the temporal, parietal and prefrontal areas (mean t-value = 2.91, p=0.002) (*Figure 2B*), the peak location of stimulus uncertainty was highly variable across subjects and included the dorso-parietal cortex, the temporo-parietal junction and the anterior cingulate cortex (*Figure 2E*).

Finally, temporal decoders of subjects' button press (left versus right index fingers) were significant from 458 ms after stimulus onset and peaked at 604 ms (AUC = 0.85; SEM = 0.011; p<0.001). A significant cluster of motor signals could be detected around sensorimotor cortices between 590 and 840 ms in the univariate analysis (mean t-value=4.98, p<0.001, *Figure 2A*). Spatial decoding corroborated this result (mean t-value 4.1, p<0.0001). Response-locked analyses revealed qualitatively similar but stronger results. For example, temporal decoders were significant from 350 ms prior to the response and up to 500 ms after the response reaching an AUC of 0.94 at response time (p<0.001). Response-locked decoding performance is shown in *Figure 2—figure supplement 2*.

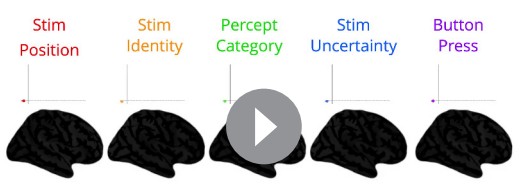

**Video 2.** Temporal decoding results. For each regressor of interest, the trajectory of normalized decoding accuracy is plot over time. The beta coefficients from the univariate spatio-temporal analysis are plot on the inflated brains, averaged over subjects. The timing of the beta coefficients corresponds to the timing of the normalized decoding accuracy, as shown in the ms counter at the bottom.
https://elifesciences.org/articles/56603#video2

Overall, the time at which representations became maximally decodable correlated with their peak location along the postero-anterior axis (*Figure 2D–E*) (r = 0.57, p<0.001). The specific hierarchical organization of the stimulus features was in some cases surprising. For example, the letter/digit contrast peaked remarkably late (~400 ms). Furthermore, Uncertainty was one of the latest features to come online, and extended into responses to the subsequent trial. These results thus strengthen the classic notion that perceptual processes are hierarchically organized across space, time and function. Importantly, however, this cascade of representations spreads over more than 600 ms and largely exceeds the time it takes the feedforward response to ignite the ventral and dorsal pathways (*Figure 1G* and *Video 1*).

## A hierarchy of recurrent layers explains the spatio-temporal dynamics of neural representations

The above results show that the brain sequentially generates, over an extended time period, a hierarchy of representations that ultimately account for perceptual reports.

To understand how this cascade of representations emanates from brain activity, we formalize four neural architectures compatible with the above results, and which nonetheless predict distinct spatio-temporal dynamics of each representation (*Figure 3*). In these models, we assume that each 'layer' generates new hierarchical features, in order to account for the organization of spatial decoders (*Figure 2E*). Furthermore, we only discuss architectures which can code for all representations simultaneously, in order to account for the overlapping temporal decoding scores (*Figure 2C*). Finally, we only model discrete activations (i.e. a representation is either encoded or not) as any more subtle variation can be trivially accounted for by signal-to-noise ratio considerations.

Each model predicts (1) 'source' decoding time courses (i.e. what is decodable within each layer) and (2) 'temporal generalization' (TG) maps. TG is used to characterize the dynamics of neural representations and consists in assessing the extent to which a temporal decoder trained at a given time sample generalizes to other time samples (*King and Dehaene, 2014b*; *Figure 3D*).

Our spatial and temporal decoding results can be accounted for by a feedforward architecture that both (i) generates new representations at each layer and (ii) propagates low-level representations across layers (*Figure 3* Model 1: 'broadcast'). This architecture predicts that representations would not be maintained within brain areas. However, this lack of maintenance is not supported by additional analyses of our data. First, the position of the stimulus was decodable in the early visual cortex between 80 and 320 ms (mean t-value=5.18, p<0.001) and thus longer than the stimulus presentation. Second, most temporal decoders significantly generalized over several hundreds of milliseconds (*Figure 4A–B*). For example, the temporal decoder trained to predict stimulus position from t = 100 ms could accurately generalize until ~500 ms as assessed with spatio-temporal cluster tests across subjects (*Figure 4A*). Similarly, temporal decoders of perceptual category and button-press generalized, on average, for 287 ms (SEM = 12.47; p<0.001) and 689 ms (SEM = 30.94; p<0.001), respectively. Given that the neural activity underlying the decoded representations is partially stable over several hundreds of milliseconds, recurrent connections seem necessary to account our data (*Figure 4* Model 2–4).

Consequently, we then considered a simple hierarchy of recurrent layers, where recurrence only maintains activated units (*Figure 3* Model 2: 'maintain'). This architecture predicts strictly square TG matrices (i.e. temporal decoders would be equivalent to one another in terms of their performance) and is thus at odds with the largely diagonal TG matrices observed empirically (*Figure 4A*). Specifically, the duration of significant temporal decoding (fitting a new decoder at each time sample) was significantly longer than the generalization of a single decoder to subsequent time samples (e.g. 1239 versus 287 ms for perceptual category (t = −61.39; p<0.001) and 1215 versus 689 ms for button-press (t = −16.26; p<0.001), *Figure 4B*). These results thus suggest that the decoded representations depend on dynamically changing activity: that is each feature is linearly coded by partially distinct brain activity patterns at different time samples.

It is difficult to determine, with MEG alone, whether such dynamic maintenance results from a change of neural activity within or across brain areas. Indeed, Model 1 and Model 3 can equally predict diagonal TG (*Figure 3*). However, these two models, and their combination (Model 4) diverge in terms of *where* information should be decodable. Specifically, source analyses revealed that both stimulus position and perceptual category can be decoded across a wide variety of partially-overlapping brain areas (*Figure 2B*, *Video 2*), similarly to Model 4. Nonetheless, our MEG study remains

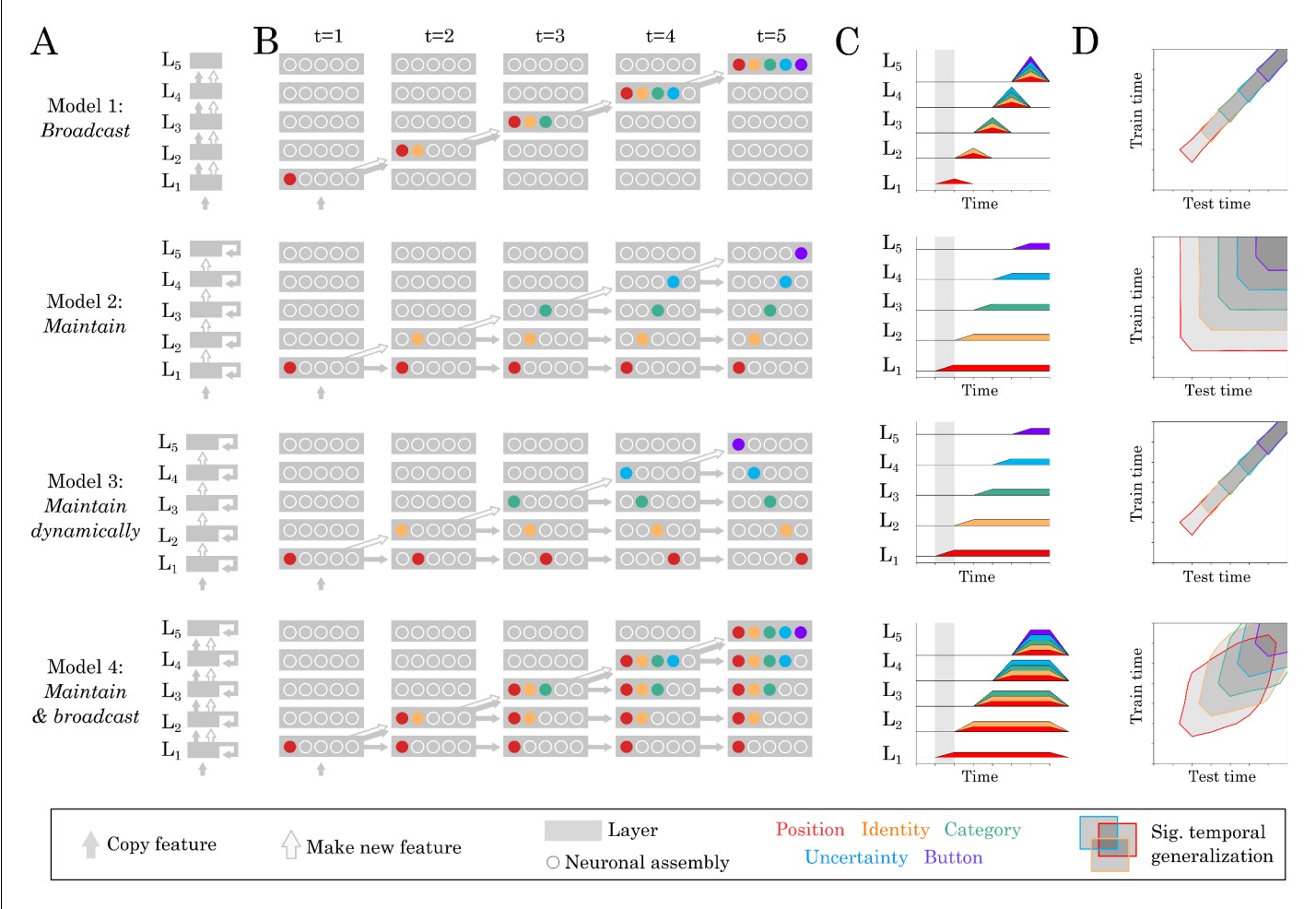

**Figure 3.** Source and temporal generalization predictions for various neural architectures. (**A**) Four increasingly complex neural architectures compatible with the spatial and temporal decoders of *Figure 2*. For each model (rows), the five layers (L1, L2 . . . L5) generates new representations. The models differ in their ability to (i) propagate low-level representations across the hierarchy, (ii) maintain information with each layer in a stable or dynamic way. (**B**) Activations within each layer plotted at five distinct time samples. Dot slots indicate different neural assemblies within the same layer. Colors indicate which feature is linearly represented. For clarity purposes, only effective connections are plotted between different time samples. (**C**) Summary of the information represented within each layer across time. (**D**) Expected result for of the temporal generalization analyses, based on the processing dynamics of each model.

limited in assessing whether within brain regions dynamics also contribute to the diagonal TG, which would suggest a mixture between models 3 and 4.

Together, source and TG analyses thus suggest that the slow and sequential generation of increasingly abstract representations depends on a hierarchy of recurrent layers that generate, maintain and broadcast representations across the cortex.

## Hierarchical recurrence induces an accumulation of delays

Can a hierarchy of recurrent processes account for the variation in single-trial dynamics? To address this issue, we hypothesized that recurrent processes would take variable amounts of time to converge to each intermediary representation. In this view, (i) each feature is predicted to propagate across brain areas at distinct moments, and (ii) the successive rise of decodable representations is thus predicted to incrementally correlate with reaction times (*Figure 5A–E*).

To test this hypothesis, we estimated how the peak of each temporal decoder varied with reaction times. For clarity purposes, we split reaction times into four quantiles, and averaged the time courses of temporal decoders relative to their training time. This method allowed us to assess the

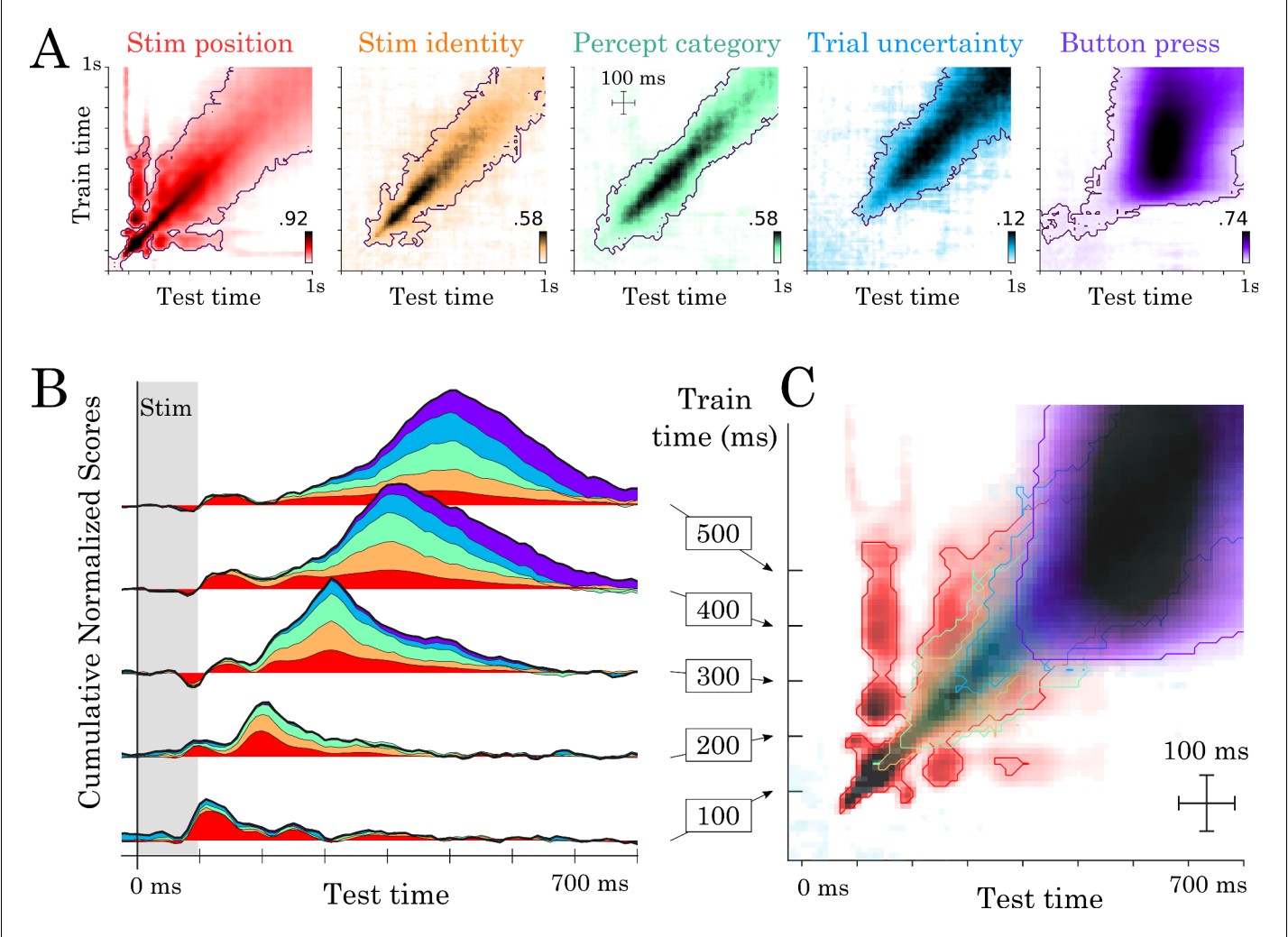

**Figure 4.** Temporal generalization results. (A) Temporal generalization for each of the five features orthogonally varying in our study. Color indicate decoding score (white = chance). Contours indicate significant decoding clusters across subjects. (B) Cumulative temporal generalization scores for the temporal decoders trained at 100, 200, 300, 400 and 500 ms, respectively. These decoding scores are normalized by mean decoding peak for clarity purposes. (C) Same data as A but overlaid. For clarity purposes, contours highlight the 25th percentile of decoding performance.

extent to which neural processes are 'sped up' or 'slowed down' relative to the average processing speed, as represented by the diagonal axis (*Figure 5—figure supplement 1* summarizes this method). These analyses showed that the latencies of (i) perceptual category (r = 0.35; p=0.006), (ii) stimulus uncertainty (r = 0.37; p=0.004) and (iii) button press (r = 0.66; p<0.001) increasingly correlated with reaction times (*Figure 5F–G*).

Overall, these results show that we can track with MEG, a series of decisions generated by hierarchical recurrent processes. This neural architecture partially accounts for subjects' variable and relatively slow reaction times.

## Hierarchical recurrence implements a series of all-or-none decisions

An architecture based on successive decisions predicts a loss of ambiguous information akin to all-or-none categorization across successive processing stages (*Figure 6A*). To test this prediction, we quantified the extent to which the decoding of 'percept category' and of 'motor action' varied linearly or categorically with (i) categorical evidence and (ii) motor evidence respectively (i.e. the extent to which the stimulus (i) objectively looks like a letter or a digit and (ii) should have led to a left or right button press given its pixels). Note that due to the limitations of our experimental design, we can only assess the effect of stimulus evidence at these two level of representations: stimulus

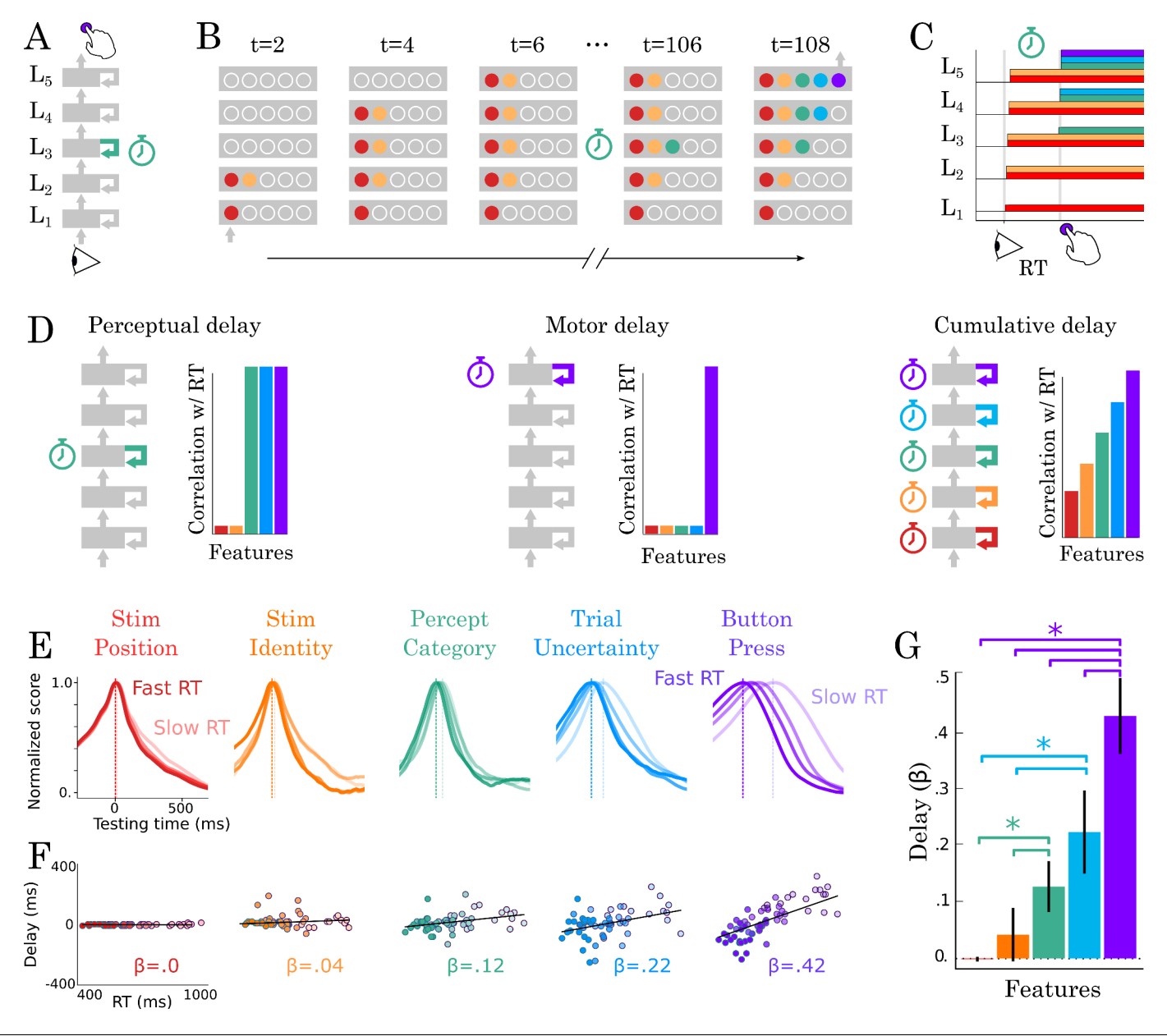

**Figure 5.** Correlation between TG peaks and reaction times. (A, B) Recurrent processing at a given processing stage is hypothesized to take a variable amount of time to generate adequate representations. (C) According to this hypothesis, the rise of the corresponding and subsequent representations would correlate with reaction times. (D, left) Predictions when delays are only induced by the perceptual stage of processing. (D, middle) Predictions when delays are only induced by the motor processing processing stage. (D, right) Predictions when delays are induced by all processing stages. (E) TG scores aligned to training time, split into trials within the fastest and slowest reaction-time quantile and averaged across reaction times bins. Dark and light lines indicate the average decoding performance for trials with fastest and slowest reaction times respectively. (F) Each subject (dot) mean peak decoding time (y-axis) as a function of reaction time (x-axis) color-coded from dark (fastest) to light (slowest). The beta coefficients indicate the average delay estimate. (G) The average slope between processing delay and reaction time for each feature. Error-bars indicate the SEM.

The online version of this article includes the following figure supplement(s) for figure 5:

**Figure supplement 1.** Schematic of the processing delay analysis pipeline.

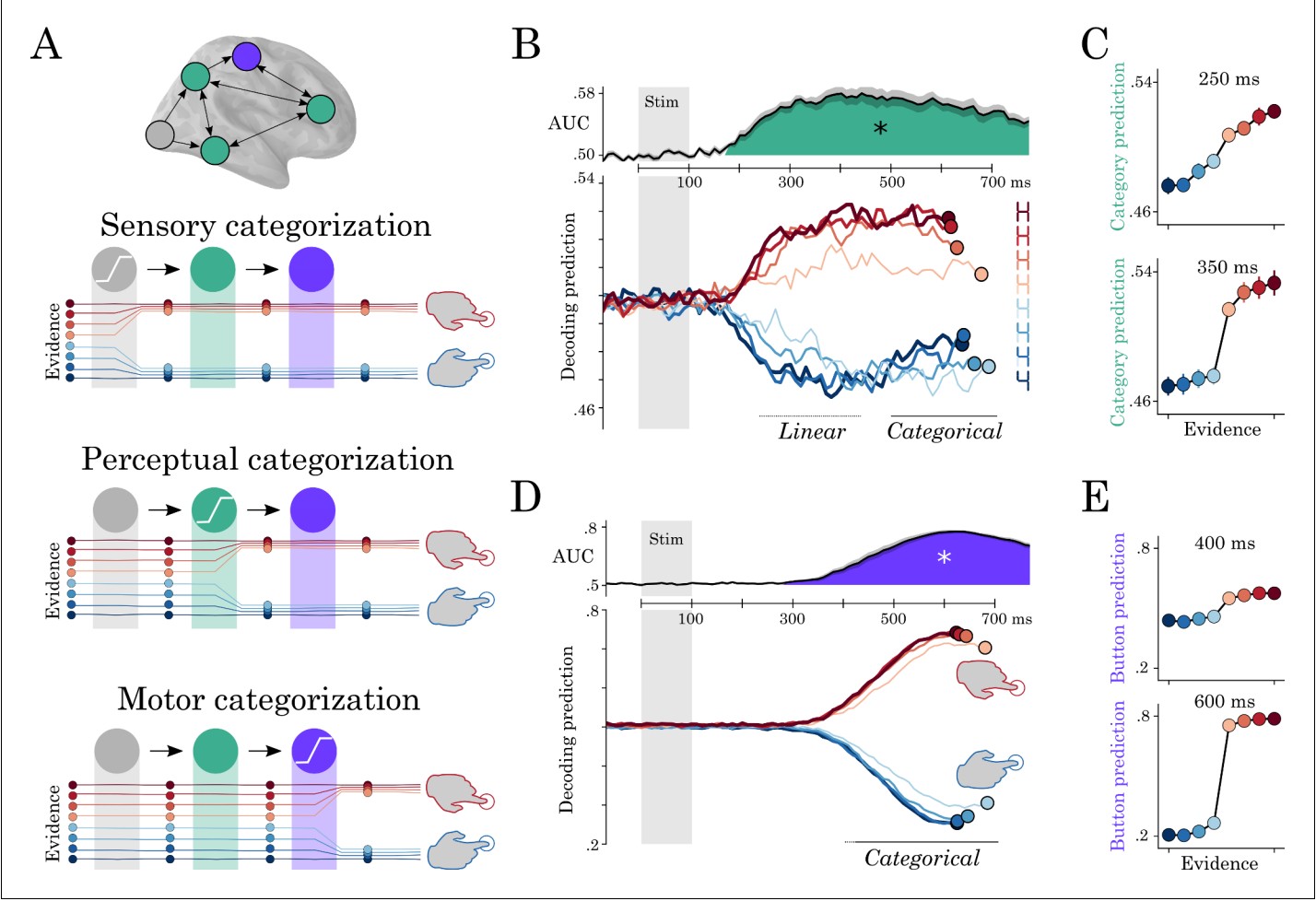

**Figure 6.** Motor and perceptual decisions. (A) Hypothesis space for when responses become categorical: during sensory, perceptual or motor processing. (B, top) Time course of decoding the perceptual decision. (B, bottom) Classifier predictions split into different levels of sensory evidence. (C) Averaging probabilities in different time-windows shows the linear-categorical shift in how information is represented. (D, top) Time course of decoding the motor decision. (D, bottom) Splitting classifier predictions into different levels of uncertainty. (E) Different windows of classifier predictions, showing the categorical responses throughout processing.

The online version of this article includes the following figure supplement(s) for figure 6:

**Figure supplement 1.** Decoding the distinction between the unambiguous end-points and stimuli at different step distances away.

**Figure supplement 2.** Decoding performance for the letter/digit contrast in active trials requiring a button press (blue line), and passively presented non-ambiguous trials requiring no response (red line).

evidence only varied as a function of decision and motor response but was orthogonal to stimulus position and stimulus identity.

The probabilistic decoding predictions of percept category correlated linearly with sensory evidence between 210 and 530 ms (r = 0.38 ±0.03, temporal-cluster p<0.001). The spatial decoders fit from 200 to 400 ms clustered around the VWFA (mean t-value=4.6; p=0.02; 224 vertices). These results suggest that this region first represents the stimulus objectively (i.e. in its full ambiguity).

Between 400 and 810 ms, the predictions of 'perceptual category' decoders were better accounted for by sigmoidal (r = 0.77 ±0.03, p<0.001) than by linear trends (r = 0.7 ±0.03, p<0.001). This suggests that later responses track the categorical perception rather than the linearly varying input. Spatial decoding analyses restricted to the 500–700 ms time window was more distributed (mean t-value=4.4; p=0.022; 110 vertices). Finally, ambiguous stimuli (steps 5 and 6 on the continuum) reached maximum decodability 205 ms later than unambiguous stimuli (steps 1 and 8)

(p<0.001) (*Figure 6B*). The interaction between trend (linear or sigmoidal) and window latency was significant across subjects (r = 0.07; SEM = 0.01; p=0.002).

This progressive categorization of the letter/digit representations contrasts with the all-or-none pattern of motor signals. Specifically, the probabilistic predictions of button-press decoders varied categorically with response evidence from 440 to 1290 ms (sigmoid > linear cluster, mean t-value=3.17; p<0.001). There was also a more transient linear trend from 410 to 580 ms (mean t-value=3.69; p<0.001). This suggests that, unlike perceptual category, motor signals largely derive from categorical inputs.

Together, delay (*Figure 5*) and categorization (*Figure 6*) analyses thus show that perceptual representations slowly become categorical and are subsequently followed by all-or-none motor representations.

## Discussion

While the role of feedforward processes is becoming increasingly understood, *what* recurrent processes represent and *how* they are orchestrated remains largely unknown. Here, we show with source-localized MEG that recurrent processes sequentially generate, over an extended time period, a hierarchy of representations that ultimately account for the timing and the content of perceptual reports.

The conclusions of the present study are limited by two main aspects. First, while our sensory-motor and letter/digit representations are largely consistent with previous findings (*Cohen et al., 2000*; *Shum et al., 2013*), MEG source reconstruction remains imperfect. Consequently, identifying (1) the role of subcortical areas and (2) the extent to which representations dynamically change within each brain area will necessitate invasive brain recordings.

Second, our study focuses on the brain responses to individual characters. This unusual task (*King and Dehaene, 2014a*) thus adds to the long list of arbitrary stimuli used to probe the neural bases of decisions. Indeed, perceptual decisions have been investigated through the manipulation of Gabor patches (e.g. *Wyart et al., 2012*), clouds of moving dots (*Shadlen and Newsome, 2001*) and even bathroom and kitchen images (*Linsley and MacEvoy, 2014*). Our results are consistent with these studies in that perceptual decisions are represented up to the fronto-parietal cortices. In particular, *Freedman et al., 2002* parametrically manipulated images of cats and dogs through 3D morphing and also show that the lateral prefrontal cortex reflects the category of the stimuli independently of their physical similarity. The benefit of our design choice is that it allowed us to (1) orthogonalize and parametrically manipulate five levels of representations and (2) track the interplay between these different levels of representations in both time and space. In the future, it will thus be critical to verify that these findings can be observed across a wide variety of stimuli (e.g. *Kar et al., 2019*; *Kietzmann et al., 2019*), and to further investigate whether decisional boundaries can be manipulated online by specific task demands (*Freedman et al., 2002*).

Overall, our results bridge three important lines of research on the neural and computational bases of visual processing.

First, core-object recognition research, generally based on ~100 ms-long image presentations has repeatedly shown that the spiking responses of the inferotemporal cortex is better explained by recurrent models than by feedforward ones (*Lamme and Roelfsema, 2000*; *Kar et al., 2019*). In particular, *Kar et al., 2019* have recently shown that images that are challenging to recognize, lead to delayed content-specific spiking activity in the macaque's infero-temporal cortex. Similar evidence for recurrent processes was recently found using MEG (*Kietzmann et al., 2019*). Our findings, based on simpler but highly controlled stimuli, are consistent with these results and further highlight that perceptual representations are not confined to the inferotemporal cortices, but also reach a large variety of parietal and prefrontal areas (*Freedman and Miller, 2008*).

The specific order that perceptual representations were generated was not entirely predictable a priori. In particular, we were surprised to find that Uncertainty was one of the last variables to come online (~300 ms) and extended into the processing of the subsequent trial. This result may relate to the fact that, here, Uncertainty is confounded with memory and task-engagement effects, rather than solely the processing of the stimulus property per se (*Bate et al., 1998*). However, this result starkly contrasts with recent work showing very early sensitivity to Uncertainty (~50 ms) in an auditory

syllable categorization task (*Gwilliams et al., 2018*), suggesting that the latency of this response may also depend on the sensory modality and familiarity with the visual or auditory object at study.

Second, the present study makes important contributions to the perceptual decision making literature (*Gold and Shadlen, 2007*; *O'Connell et al., 2012*). With some notable exceptions (e.g. *Philiastides and Sajda, 2007*), this line of research primarily aims to isolate motor and supra-modal decision signals in the presence of sustained visual inputs: that is, neural responses ramping toward a virtual decision threshold, independently of the representation on which this decision is based (*O'Connell et al., 2012*). The present study complements this approach by tracking the representation-specific signals that slowly emerge after a brief stimulus.

Our results thus open an exciting avenue for querying the gating mechanisms of successive decisions and clarifying the role of the prefrontal areas in the coordination multiple perceptual and supramodal modules (*Sarafyazd and Jazayeri, 2019*).

Finally, our results constitute an important confirmation of modern theories of perception. In particular, the Global Neuronal Workspace Theory predicts that perceptual representations need to be broadcast to associative cortices via the fronto-parietal areas to lead to subjective reports (*Dehaene and Changeux, 2011a*). Yet, at some notable exceptions (*Tong et al., 1998*; *King et al., 2016*), previous studies often fail to dissociate perceptual contents and perceptual reports (e.g. *Sergent et al., 2005*; *van Vugt et al., 2018*). By contrast, the present experimental design allows an unprecedented dissection of the distinct processing stages that transform sensory input into perceptual representations and, ultimately, actions. The generation of letter and digit representations in the dedicated brain areas (*Cohen et al., 2000*; *Shum et al., 2013*) and their subsequent broadcast to the cortex reinforce the notion that subjective perception relate to the global sharing of content-specific representations across brain areas (*Dehaene and Changeux, 2011a*; *Lamme, 2003*).

## Materials and methods

### Target stimuli

Using the font designed in *King and Dehaene, 2014a*, the stimuli were made from 0, 4, 5, 6, 8, 9, A, C, E, H, O, S, or from a linear combination of two of these characters varying in a single black bar (hereafter 'pixel'). The corresponding 'morphs' were created by adjusting the contrast of the remaining pixel along eight equally spaced steps between 0 (no bar) and 1 (black bar).

### Experiment 1

Eight subjects with normal or corrected vision, seated ~60 cm from a 19' CRT monitor (60 Hz refresh rate, resolution: 1024 × 768), performed a stimulus identification task with continuous judgements across 28 variably ambiguous stimuli generated from digit stimuli. Ten euros were provided in compensation for this 1 hr experiment.

Subjects performed four blocks of 50 trials, each organized in the following way. After a 200 ms fixation, a target stimulus, randomly selected from one of the 28 stimuli, was flashed for 83 ms on a 50% gray background to the left or to the right of fixation. The orientation of the reporting disk (e.g 5-6-8-9 versus 5-9-8-6) was counterbalanced across subjects. Subjects had then up to 10 s to move a cursor on a large disk to report their percepts. The radius on the disk indicated subjective visibility (center = did not see the stimulus, disk border = max visibility). The angle on the disk indicated subjective identity (e.g. 5, 6, 8, 9 for the top left, top right, bottom right, and bottom left 'corners', respectively). Inter-trial interval was 500 ms. To verify that subjects provided meaningful reports, the target stimulus was absent 15% of the trials. Absent trials were rated with a low visibility (defined as radius below 5% of the disk radius) in most cases. Absent trials and trials reported with a low visibility were excluded from subsequent analyses (16% ± 1.4%). The report distribution plotted in *Figure 1B* were generated with Seaborn's bivariate Gaussian kernel density estimate function with default parameters.

#### Modeling categorical reports

To test whether subjective reports of stimulus identity varied linearly or categorically with sensory evidence, we analyzed how reports' angle (i.e. subjective identity) varied with the expected angle given the stimulus (i.e. sensory evidence).

For each morph (5–6, 5–8, 9–8 and 6–8) separately, we fit a linear model:

$$\hat{y} \leftarrow \beta_1 x + \beta_0 \tag{1}$$

and a sigmoidal model:

$$\hat{y} \leftarrow \frac{1}{1 + \exp(\beta_1 x + \beta_2)} + \beta_0 \tag{2}$$

where $\hat{y}$ is the report angle predicted by the model, $x$ is expected angle given the stimulus pixels and $\beta_0$ is a free bias parameter.

To minimize the effects of noise, behavioral reports were first averaged within each level of evidence, sorted from the stimulus with the least pixels (e.g. 5, in 5–6 morph) to the stimulus with the most pixels (e.g. 6 in the 5–6 morph). The resulting averages were normalized to range between 0 and 1 within each subject. The β parameters were fit with Scipy's 'curve_fit' function (*Jones et al., 2001*) to minimize a mean squared error across trials *i*:

$$\underset{\beta}{\arg\min} \sum_i (y_i - \hat{y}_i)^2 \tag{3}$$

Because the linear and sigmoidal models have distinct numbers of free parameters, we compared them within a five-split cross-validation. Specifically, the two models were repeatedly fit and tested on independent trials. A Pearson correlation coefficient *r* summarized the ability of each model to accurately predict $y_{test}$ given $x_{train}$, $y_{train}$ and $x_{test}$. Finally, a Wilcoxon test was applied across subjects to test whether the two models were consistently above chance ($r>0$) and consistently different from one another ($r_{sigmoid}>r_{linear}$).

## Experiment 2

This experiment was performed at Neurospin, Gif usr Yvette, thanks to the support of Stanislas Dehaene. Seventeen subjects performed a discrete identification task across 22 variably ambiguous stimuli generated from letters and digits inside an Elekta Neuromag MEG scanner (204 planar gradiometers and 102 magnetometers). Seventy euros were provided in compensation to the 1 hr experiment and 30 min of preparation.

A sample size of 17 participants was selected based on previous visual studies utilizing the same MEG machine (*King et al., 2016*).

Participants' head shape was digitized along with five fiducial points on the forehead and on each aural canal. Five head-position coils were placed on subjects head and localized at the beginning of each block.

The trial structure was as follows. A black fixation cross was displayed on a 50% gray background for 300 ms followed by a 100ms-long target stimulus presented on the left or on the right of fixation. Two task-irrelevant flankers (e.g. stimulus can be read as an S or a 5) were displayed on the side of this target stimulus to increase our chances of eliciting recurrent processing via crowding (*Strasburger et al., 2011*). Subjects were given two seconds to report the identity of the stimulus. Reports of stimulus identity were given by pressing a button with the left and right index fingers respectively. The identity-button mapping changed on every block to orthogonalize the neural correlates of stimulus identity and the neural correlates of motor actions. For example, in block 1, perceiving an E or a 4 should have been reported with a left button press, whereas in block 2, E and 4 should have been reported with a right button press. The identity-button was explicitly reminded before each block. In addition, a visual feedback was displayed after non-ambiguous trials. Specifically, the fixation turned green for 100 ms or red for 300 ms in correct and incorrect trials respectively. The brain responses to these feedback stimulations are not analyzed in the present study. Inter-trial interval was 1 s. Subjects were provided a short training to ensure they understood the task, and identified non-ambiguous targets at least 80% of the time.

A total of 1920 trials, grouped into 40 blocks, were performed by each subject, 320 of which were presented passively at the end of each block – subjects were not required to provide a response. The trial structure was generated by (i) permuting all combinations of stimulus features (e.g. position, identity, response mapping, uncertainty), and (ii) shuffling the order of presentation for each subject. The experiment was presented using Psychtoolbox (*Kleiner et al., 2007*).

All experiments were approved by the local ethics committee. All subjects signed an informed consent form.

## Structural MRI

For each subject, an anatomical MRI with a resolution of $1 \times 1 \times 1.1$ mm was acquired after the MEG experiment with a 3T Siemens scanner. Gray and white matter were segmented with Freesurfer 'recon-all' pipeline (*Fischl, 2012*) and coregistered with each subject's digitized head shapes along with fiducial points.

## Preprocessing

The continuous MEG recording was noise-reduced using Maxfilter's SSS correction on the raw data, bandpass-filtered between 0.5 and 40 Hz using MNE-Python's default parameters with firwin design (*Gramfort et al., 2014*) and downsampled to 250 Hz. Epochs were then segmented between $-300$ ms and $+1500$ ms relative to stimulus onsets.

After coregistering the MEG sensor data with subjects' structural MRI and the head position coils, we computed the forward model using a 1-layer (inner skull) boundary element model, for each subject separately and fit a minimum-norm inverse model (signal to noise ratio: 3, loose dipole fitting: 0.2, with normal orientation of the dipole relative to the cortical sheet) using the noise covariance across sensors averaged over the pre-stimulus baseline across trials. Finally, the inverse model was applied to single-trial data resulting in a dynamic Statistical Parameter Map (dSPM) (*Dale et al., 2000*) value for each source at each time sample.

## Modeled features

We investigated whether single-trial source and sensor evoked responses varied as a function of five features: (1) the position of the stimulus on the computer screen (left versus right of fixation), (2) the morph from which the stimulus is generated (E-6 versus H-4), (3) the category of the stimulus (letter versus digit), (4) the uncertainty of the trial (maximum uncertainty = stimuli with pixel at 50% contrast; minimally uncertain stimuli with pixels at 0% or 100% contrast), and (5) the response button used to report the stimulus (left versus right button). By design, these five features are independent of one another.

It is challenging to dissociate brain responses that represent objective sensory information from those that represent perceptual decisions as the two are generally collinear. To address this issue, we first fit univariate and multivariate models to predict perceptual category: that is, whether the button press indicated a character that belongs to the digit or to the letter category. This feature is independent of the button press (e.g. the letter E and the digit 4 can be reported with the same button in a given block). Furthermore, this feature is not necessary to perform the task (i.e. knowing whether E and H are letters is unnecessary to discriminate them). We reasoned that if subjects automatically generates letter/digit representations during perceptual categorization, then we should be able to track the generation of this abstract feature from brain activity.

## Mass univariate statistics

To estimate whether brain responses correlated with each of these five features, we first fit, within each subject, mass univariate analyses at each source location and for each time sample with a linear regression:

$$\beta = (X^T X)^{-1} X y \tag{4}$$

where $X \in \mathbb{R}^{n,f}$ is a design matrix of $n$ epochs by $f = 5$ features and $y \in \mathbb{R}^n$ is the univariate brain response at a given source and at given time. The effect sizes β were then passed to second-level statistics across subjects corrected for multiple comparisons using non-parametric spatio-temporal cluster testing (see below).

## Decoding

Decoding analyses consists in predicting each feature from multivariate brain responses. Decoding analyses were performed within a five-split stratified K-Fold cross-validation using l2-regularized

linear models. Classifiers consisted of logistic regressions (with scikit-learn *Pedregosa, 2011*'s default parameters: $C = 1$):

$$\underset{\beta}{\mathrm{argmin}} \sum_i log(1 + exp(-y_i \beta^T \vec{x}_i)) + C\|\beta\|^2 \qquad (5)$$

where $y_i \in \{\pm 1\}$ is the feature to be decoded at trial $i$ and $x_i$ is the multivariate brain response.

Regressors consisted of ridge regression (with scikit-learn *Pedregosa, 2011*'s default parameters: $\alpha = 1$).

$$\underset{\beta}{\mathrm{argmin}} \sum_i (y_i - \beta^T x_i)^2 + \alpha\|\beta\|^2 \qquad (6)$$

For each subject independently, decoding performance was summarized across trials, with an area under the curve (AUC) and a Spearman $r$ correlation score for classifiers and regressors, respectively.

All decoders were provided with data normalized by the mean and the standard deviation in the training set.

Spatial decoding consists in fitting a series of decoders at each brain source independently, across all 1500 time samples relative to stimulus onset. This analysis results in a decoding brain map that indicates where a feature can be linearly decoded in the brain. These decoding maps were then passed to cluster-corrected second-level statistics across subjects.

Temporal decoding consists in fitting a series of decoders at each time sample independently, across all 306 MEG sensors. This analysis results in a decoding time course that indicates when a feature can be linearly decoded from MEG signals. These decoding time courses were then passed to cluster-corrected second-level statistics across subjects.

Temporal generalization (TG) consists in testing whether a temporal decoder fit on a training set at time $t$ can decode a testing set at time $t'$ (*King and Dehaene, 2014b*). TG can be summarized with a square training time × testing time decoding matrix. To quantify the stability of neural representations, we measured the duration of above-chance generalization of each temporal decoder. To quantify the dynamics of neural representations, we compared the mean duration of above-chance generalization across temporal decoders to the duration of above-chance temporal decoding (i.e. the diagonal of the matrix versus its rows). These two metrics were assessed within each subject and tested with second-level statistics across subjects.

## Permutation cluster test

To evaluate the statistical significance of the univariate and multivariate analyses, we used a one-sample permutation cluster test as implemented in MNE-Python (*Gramfort et al., 2014*). We use the default parameters of the 'spatio temporal cluster one sample test' from mne version 0.17.1.

First we center the data around the theoretical chance level (e.g. 0.5 for AUC, 0 for Spearman correlation or beta coefficient). A one-sample t-test is performed at each location in time and space. Then, spatio-temporally adjacent data-points are clustered based on a cluster-forming threshold of p<0.05. The test statistic for each cluster is the sum of the t-values across time and space. Randomized data are generated with random sign flips, and a new set of clusters are formed. The null distribution is created based on the summed t-values that are generated from 5000 random permutations of the data. This analysis follows *Maris and Oostenveld, 2007*.

## Linear versus categorical

To test whether neural representations varied as a function of (i) reaction times (RTs, split into four quantiles), (ii) sensory evidence (i.e. the extent to which the stimulus objectively corresponds to a letter) and (iii) motor evidence (i.e. whether the stimulus should have led to the left button press), we analyzed the extent to which decoders' predictions covaried with each of these three variables $z$:

$$f(z, \beta^T X) \qquad (7)$$

where $f$ is a linear or a sigmoidal model, $X$ is the multivariate brain response and β is the decoder's coefficient fit with cross-validation.

## Statistics

Univariate, decoding and TG models were fit within subjects, and tested across subjects. In case of repeated estimates (e.g. temporal decoding is repeated at each time sample), statistics derived from non-parametric cluster-testing with 10,000 permutations across subjects with MNE-Python's default parameters (*Gramfort et al., 2014*).

### Simulations

To formalize how distinct neural architectures lead to distinct spatio-temporal dynamics, we modeled discrete linear dynamical systems forced with a transient input *U*. Specifically:

$$X_{t+1} = AX_t + BU_t \tag{8}$$

where X is a multidimensional times series (i.e. neurons x time), A is the architecture, and corresponds to square connectivity matrix (i.e. neurons x neurons), B is an input connectivity matrix (i.e. inputs x neurons), and U is the input vector.

Distinct architectures differ in the way units are connected with one another. For simplicity purposes, we order units in the *A* matrix such that their row index correspond to their hierarchical levels.

In this view, the recurrent, feedforward and skip connections of the architecture A were modeled as a binary diagonal matrix *R*, a shift matrix *F* and a matrix *S* with one entries in the last column respectively. These three matrices were modulated by specific weights, as detailed below. The input *U* was only connected to the first 'processing stage', that is, to the first unit(s) of *A*, via a matrix *B* constant across architectures, and consisted of a transient square-wave input, that mimics the transient flash of the stimulus onto subjects' retina.

To model multiple features, we adopted the same procedure with multiple units per layer. Each unit within each layer was forced to encode a specific feature.

Each architecture shown in *Figure 3* was fed an input at t = 1, and simulated for eight time steps. Finally, temporal generalization analyses based on the architectures' activations were applied for each of the features.

The same architecture is shown in *Figure 5B*. Here we simulate for a total of 108 time-steps, with an arbitrary delay of 100 time-steps between t = 6 and t = 106.

## Acknowledgements

This project received funding from the European Union's Horizon 2020 research and innovation program under grant agreement No 660086, the Bettencourt-Schueller Foundation, the Fyssen Foundation, the Philippe Foundation, the Abu Dhabi Institute G1001, NIH R01DC05660 and the Dingwall Foundation. We are infinitely grateful to Stanislas Dehaene, as well as David Poeppel and Alec Marantz, for their support. We thank Michael Landy for his very helpful and generous feedback on a previous version of the manuscript.

## Additional information

### Funding

| Funder | Grant reference number | Author |
|--------|------------------------|--------|
| William Orr Dingwall Foundation | Dissertation Fellowship | Laura Gwilliams |
| Abu Dhabi Institute Grant | G1001 | Laura Gwilliams |
| Horizon 2020 Framework Programme | 660086 | Jean-Remi King |
| Fondation Bettencourt Schueller | Bettencourt-Schueller Foundation | Jean-Remi King |
| Fondation Roger de Spoelberch | Fondation Roger de Spoelberch | Jean-Remi King |

| Philippe Foundation | Philippe Foundation | Jean-Remi King |
| National Institutes of Health | R01DC05660 | Laura Gwilliams |

The funders had no role in study design, data collection and interpretation, or the decision to submit the work for publication.

### Author contributions

Laura Gwilliams, Conceptualization, Formal analysis, Investigation, Visualization, Writing - original draft, Writing - review and editing; Jean-Remi King, Conceptualization, Formal analysis, Supervision, Investigation, Visualization, Methodology, Writing - original draft, Writing - review and editing

### Author ORCIDs

Laura Gwilliams https://orcid.org/0000-0002-9213-588X
Jean-Remi King http://orcid.org/0000-0002-2121-170X

### Ethics

Human subjects: This study was ethically approved by the comité de protection des personnes (CPP) IDF 7 under the reference CPP 08 021. All subjects gave written informed consent to participate in this study, which was approved by the local Ethics Committee, in accordance with the Declaration of Helsinki. Participants were compensated for their participation.

### Decision letter and Author response

Decision letter https://doi.org/10.7554/eLife.56603.sa1
Author response https://doi.org/10.7554/eLife.56603.sa2

## Additional files

### Supplementary files

• Supplementary file 1. Summary table showing the timing and significance of the results across the five features across the three statistical analyses (temporal decoding, spatial decoding and mass uni-variate). pfc corresponds to the p-value after Bonferroni correction across the five features. Average t-value corresponds to the average t-value in the cluster.

• Transparent reporting form

### Data availability

Anonymised source data for figures have been uploaded to Dryad: https://doi.org/10.5061/dryad.70rxwdbtr.

The following dataset was generated:

| Author(s) | Year | Dataset title | Dataset URL | Database and Identifier |
|---|---|---|---|---|
| Gwilliams L, King J-R | 2020 | Data from: Recurrent processes support a cascade of hierarchical decisions | https://doi.org/10.5061/dryad.70rxwdbtr | Dryad Digital Repository, 10.5061/dryad.70rxwdbtr |

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
