## [Decision Letter]

Thank you for submitting your article "Recurrent processes support a cascade of hierarchical decisions" for consideration by *eLife*. Your article has been reviewed by three peer reviewers, and the evaluation has been overseen by Thomas Serre as Reviewing Editor and Michael Frank as the Senior Editor. The reviewers have opted to remain anonymous.

The reviewers have discussed the reviews with one another and the Reviewing Editor has drafted this decision to help you prepare a revised submission.

Summary:

The authors use a combination of MEG, structural MRI, and computational modeling to measure how the visual cortex accumulates information for discriminating between objects. Using a digit/letter warping dataset, the authors identify difficult exemplars, devise clever analyses to show when representations become categorical, and combine temporal decoding analyses with modeling to describe dynamics and infer computations. This paper has a significant number of results, elegantly presented in beautiful figures. While some (if not most) of the conclusions derived from the work may have been reached independently by prior studies, one major strength of the present manuscript is to examine all these questions within a single dataset. Further qualities are the use of an elegant experimental design, thorough decoding analysis methods, and adequate use of modeling to help disentangle alternative explanations.

However, as detailed below, the reviewers have identified a number of weaknesses and are requesting that the authors comment on these critiques. One of the main issues raised by the reviewers has to do with some of the effect sizes and underlying statistical tests.

Essential revisions:

Statistics

The reviewers struggled a bit with the effect sizes reported. The authors normalize their scores (Figure 2C, Figure 4), which makes the effects look strikingly similar. But the truth is that some of the effect sizes are so small (AUCs between 0.5-0.6) that it can be hard to accept some of the findings. The reviewers would like to ask the authors to think of additional analyses that they could run that would ease these concerns.

While the maximum values of the curves in this figure are very different in range, the variations in baselines of blue, green, and orange curves do not show the scaling. Please provide figures without the scaling.

Is the trial uncertainty decoding time course significant? The effect size is very small compared to other features and it is very distributed over the brain which makes us wonder if this feature is actually readable from the brain activity.

There are no statistical tests reported in Figure 1G. Please mark significant decoding scores over time by drawing a contour line around the significant clusters. Please describe the statistical tests in Figure 2A and B as the multiple comparison corrections are unclear.

Figure 2A is thresholded based on t-values that exceed an uncorrected p <.1. The reviewers are hoping this is a typo.

For the curves in Figure 2C, the authors do not indicate the time points when the scores are above chance. For example, we do not know if the blue curve with a max of 0.08 is even significant.

In addition to multiple comparison corrections across time, the authors should correct for multiple comparisons across five features.

The authors have not reported the thresholds they use for cluster definition and cluster size corrections. Please comment. This is especially important because it is not clear if the authors have also corrected for 5 multiple comparisons across the five features.

Subsection “Hierarchical recurrence implements a series of all-or-none decisions”: Between 400 and 810 ms, the predictions of 'perceptual category' decoders were better accounted for by sigmoidal (r=0.77 +/-0.03, p<0.001) than by linear trends (r=0.77 +/-0.03, p<0.001)? Please comment.

Subsection “Hierarchical recurrence induces an accumulation of delays”: the authors test the correlation of peak latency of averaged temporal decodings when averaged over training times. Please do this analysis with the temporal decoding time courses in Figure 2C. Because the main temporal dynamics occur along the diagonal of the TG decoding matrix.

Modeling

Another issue is with the modeling simulations described in the subsection “Statistics”, which disambiguates between the hypotheses in Figure 3. The reviewers' (maybe incorrect) interpretation was that the authors tested whether or not these stimuli are being processed via hierarchical recurrent computations or not. The reviewers thought this was a strawman argument, as there is no reason to suspect the converse (non-hierarchical/non-recurrent). This modeling work thus only added to their overall feeling that the contributions of the present manuscript were actually quite limited.

Interpretation

We would suggest adding clear statements in the Abstract and in the Discussion cautioning that these exact results may be limited to this specific task (difficult digit vs. letter classification), and could differ for other tasks (e.g. simple detection task, natural scene or object categorization).

Contributions

There must be a discussion of (Freedman et al., 2002). Those authors parametrically warped dog/cat stimuli to show that a region of the prefrontal cortex (PFC) reflected stimulus discriminability. This is of course closely related to the present work, where the authors use digit/letter stimuli to accomplish the same thing and focus on the visual cortex rather than PFC. The reviewers request some clarification about the contributions of the present study in light of this work and especially discuss the possibility that most of the presented results could be reflecting common input from PFC as suggested by Linsley and MacEvoy, 2014.

[Editors' note: further revisions were suggested prior to acceptance, as described below.]

Thank you for resubmitting your article "Recurrent processes support a cascade of hierarchical decisions" for consideration by *eLife*. Your revised article has been reviewed by three peer reviewers, and the evaluation has been overseen by a Reviewing Editor and Michael Frank as the Senior Editor. The reviewers have opted to remain anonymous.

The reviewers have discussed the reviews with one another and the Reviewing Editor has drafted this decision to help you prepare a revised submission.

The reviewers made a few additional comments.

1) The most important one deals with the lack of significance in your mass-univariate analyses. We suggest you describe the analyses in the main text and explain that it did not reach significance. Because the results make sense, the reviewers suggest to keep them but to move it to the SI as they should be taken with a grain of salt (and state that).

2) Related to comment 2 on statistics, the figures in Figure 2—figure supplements 1 and 2 again are scaled; because the y-axis of all plots are scaled to the maximum of each plot.

---

## [Author Response]

Essential revisions:StatisticsThe reviewers struggled a bit with the effect sizes reported. The authors normalize their scores (Figure 2C, Figure 4), which makes the effects look strikingly similar. But the truth is that some of the effect sizes are so small (AUCs between 0.5-0.6) that it can be hard to accept some of the findings… The reviewers would like to ask the authors to think of additional analyses that they could run that would ease these concerns.

Thank you for highlighting this issue. It is true that some of the effect sizes are small. However, the vast majority of these effects are highly reliable and consistent across subjects. In addition, the motivation behind the normalization of the decoding scores is to highlight their temporal differences. We believe that the successive development of the five representations is a more valuable information than the fact that the visual position of the stimulus is much easier to decode than its letter/digit category.

To strengthen confidence in the robustness of our results we have now added:

i) Several supplementary figures, including non-scaled decoding time-courses (Figure 2—figure supplements 1 and 2), and a video showing the effects of each individual subject (violin plots) across time (Figure 2—animation 1).

ii) A number of provisos in the text itself highlighting the small effect sizes where appropriate.

We believe that these additional results make it clear that although the decoding scores obtained within each subject may be small in effect size, they are sufficiently consistent across subjects to support our conclusions.

While the maximum values of the curves in this figure are very different in range (.), the variations in baselines of blue, green, and orange curves do not show the scaling. Please provide figures without the scaling.

We agree with this remark. We have added stimulus-locked and response-locked non-scaled decoding time-courses (Figure 2—figure supplements 1 and 2).

Is the trial uncertainty decoding time course significant? The effect size is very small compared to other features and it is very distributed over the brain which makes us wonder if this feature is actually readable from the brain activity.

Yes, the uncertainty decoding is significant (Average R = 0.12; SEM = 0.024; p < 0.01) (subsection “Neural representations are functionally organized over time and space”).

Note that because this is a continuous variable, we used a ridge regression to decode it, and a Spearman correlation to evaluate the performance of this decoder. This analysis is thus in a different metric scale than the categorical variables, which are decoded with a Logistic Regression and summarized with an AUC.

We have clarified this issue in the text.

There are no statistical tests reported in Figure 1G. Please mark significant decoding scores over time by drawing a contour line around the significant clusters. Please describe the statistical tests in Figure 2A and B as the multiple comparison corrections are unclear.

We apologize for this omission. We have now added analysis details regarding the univariate spatio-temporal cluster test, and the multivariate spatial cluster test in the text. In addition, we now supply supplementary figures showing the masks of significant decoding accuracy for both tests (Figure 2—figure supplements 4-9).

For the multivariate spatial cluster test, decoding performance is only displayed for sources that are within a significant cluster (p <.05). We have made this clear in the figure legend.

Overall, these additional analyses confirm and strengthen our original results.

Figure 2A is thresholded based on t-values that exceed an uncorrected p <.1. The reviewers are hoping this is a typo.

This is not a typo, but an explicit choice, which we insufficiently explained in our original manuscript.

For the mass-univariate analyses in source-space the effect of letter/digit decision did not reach statistical significance after control for multiple comparisons (p = 0.21). The spatio-temporal test was applied across the whole brain (~5,000 sources) and across the entire epoch (0:1500 ms), making the procedure very conservative. Because the letter/digit contrast is highly significant when using temporal (p <.001) and spatial (p <.001) decoders, we reasoned it would nonetheless be informative to visualize where the corresponding univariate peak activity would be. We thus adapted the plotting threshold to display the location of the strongest univariate effects. These results point to the Visual Word Form and Number Form Area (Dehaene and Cohen, 2011), as expected.

Overall, these results illustrate that multivariate decoding analyses can be much more sensitive to subtle effects, which would have otherwise been missed with standard mass-univariate analyses, should we not know a priori where to look for them. Nonetheless, these results come at the price of a diminished spatial or temporal specificity.

We have clarified this in the text and the figure legend.

For the curves in Figure 2C, the authors do not indicate the time points when the scores are above chance. For example, we do not know if the blue curve with a max of 0.08 is even significant.

Thank you for the suggestion. We have now added indicators of significance for the temporal cluster test in Figure 2C.

In addition to multiple comparison corrections across time, the authors should correct for multiple comparisons across five features.

As described in the Materials and methods, we use a temporal and/or spatial permutation cluster test which allows us to avoid the issue of corrections for multiple comparisons across time and space.

We have not corrected our results for the five features of interest in the main text. However, we have added feature-corrected results to the supplementary materials, broken down into temporal decoding, spatial decoding and mass univariate analyses. Note that this additional correction does not influence the interpretation of any of our results.

The authors have not reported the thresholds they use for cluster definition and cluster size corrections. Please comment. This is especially important because it is not clear if the authors have also corrected for 5 multiple comparisons across the five features.

We apologize for this lack of precision. For all permutation cluster tests we use the default parameters as provided in the Python module MNE-Python version 0.17.1. Clusters are formed with an initial p <.05 threshold. No cluster size correction is applied. We have added this information to the Materials and methods section of the manuscript.

Subsection “Hierarchical recurrence implements a series of all-or-none decisions”: Between 400 and 810 ms, the predictions of 'perceptual category' decoders were better accounted for by sigmoidal (r=0.77 +/-0.03, p<0.001) than by linear trends (r=0.77 +/-0.03, p<0.001)? Please comment.

Thank you for pointing out this mistake. The linear trend was r=0.7 +/- 0.03, p<0.001, which we have now added to the manuscript.

Subsection “Hierarchical recurrence induces an accumulation of delays”: the authors test the correlation of peak latency of averaged temporal decodings when averaged over training times. Please do this analysis with the temporal decoding time courses in Figure 2C. Because the main temporal dynamics occur along the diagonal of the TG decoding matrix.

The delay analysis is not based on the diagonal decoding peak latency. We operationalize delays in processing as shifts relative to the diagonal plane. This involves first aligning the test-time axis relative to the diagonal, and then averaging over train time. To aid interpretation we have added a schematic of the method (Figure 5—figure supplement 1).

ModelingAnother issue is with the modeling simulations described in the subsection “Statistics”, which disambiguates between the hypotheses in Figure 3. The reviewers' (maybe incorrect) interpretation was that the authors tested whether or not these stimuli are being processed via hierarchical recurrent computations or not. The reviewers thought this was a strawman argument, as there is no reason to suspect the converse (non-hierarchical/non-recurrent). This modeling work thus only added to their overall feeling that the contributions of the present manuscript were actually quite limited.

All but one model included both hierarchical *and* recurrent processes. Consequently, the main goal of our modelling was not to test *whether* hierarchy and recurrency exists, but rather *how* they can be characterized and *what* underlying neural architecture their spatio-temporal dynamics imply. To clarify this issue, we added a paragraph to introduce the model comparisons and amended the discussion.

InterpretationWe would suggest adding clear statements in the Abstract and in the Discussion cautioning that these exact results may be limited to this specific task (difficult digit vs. letter classification), and could differ for other tasks (e.g. simple detection task, natural scene or object categorization).

This is a fair remark. While it applies to the vast majority of perceptual decision-making studies, we amended the Abstract and Discussion to highlight the fact that we specifically focus on the restricted case of reading individual characters.

ContributionsThere must be a discussion of (Freedman et al., 2002). Those authors parametrically warped dog/cat stimuli to show that a region of the prefrontal cortex (PFC) reflected stimulus discriminability. This is of course closely related to the present work, where the authors use digit/letter stimuli to accomplish the same thing and focus on the visual cortex rather than PFC. The reviewers request some clarification about the contributions of the present study in light of this work and especially discuss the possibility that most of the presented results could be reflecting common input from PFC as suggested by Linsley and MacEvoy, 2014.

We agree with the relevance of the study from Freedman and colleagues (Freedman et al., 2002), in which they recorded PFC neurons in monkeys performing a cats/dogs categorization task. Our study supplements this work in that we

1) estimate the neural responses of a much larger set of brain areas

2) parametrically manipulate 5 levels of representations

3) formalize the different recurrent architectures that could have accounted for our decoding scores

4) record from human subjects

We have now added a paragraph in the Discussion relating our results to those of Freedman et al., 2002, and made explicit the contribution of the current work. We also mention Linsley and MacEvoy’s paper as one of the papers investigating perceptual decision using a different set of stimuli.

[Editors' note: further revisions were suggested prior to acceptance, as described below.]

The reviewers made a few additional comments.1) The most important one deals with the lack of significance in your mass-univariate analyses. We suggest you describe the analyses in the main text and explain that it did not reach significance. Because the results make sense, the reviewers suggest to keep them but to move it to the SI as they should be taken with a grain of salt (and state that).

While we appreciate this concern given the weakness of some of the effects, we believe that it is important to keep the univariate figure for a number of reasons.

First, all but one of the analyses are significant. Removing this figure because *one* of the effects does not reach statistical significance is not legitimate.

Second, we explicitly warn the reader both in the figure legend and in the main text that one of the univariate effects (letter/digit category) does not reach statistical significance after correction for multiple comparisons:

“Modelling neural activity as a function of our orthogonal stimulus properties yielded non-significant results for the decision variable of interest. […] The direct comparison between the sensitivity of the mass-univariate versus multivariate approaches are shown in Figure 2.”

Third, keeping the univariate analyses allows us to highlight the robustness and complementarity of the multivariate analyses. Specifically, MVPA over time clearly shows that letter/digit category can be decoded from ~150 ms after word onset, and MVPA over space clearly shows that this representation peaks around the visual word form area. These significant effects are compatible with the trend that we observed with univariate analyses.

Fourth, the non-significant univariate trends actually correspond to the region of interest that we expected: i.e. the visual word form area (Dehaene and Cohen, 2011).

Together, these elements make us believe that the univariate figures should be present in the main text. We have, however, added the non-thresholded maps of p-values in supplementary materials for completeness purposes.

2) Related to comment 2 on statistics, the figures Figure 2—figure supplements 1 and 2 again are scaled; because the y-axis of all plots are scaled to the maximum of each plot.

We have now added a non-scaled version of Figure 2C in the supplementary materials, time locked to stimulus onset and response onset.